# Dietary Problems Are Associated with Frailty Status in Older People with Fewer Teeth in Japan

**DOI:** 10.3390/ijerph192316260

**Published:** 2022-12-05

**Authors:** Takashi Miyano, Ryosuke Kaneko, Toshihide Kimura, Misa Maruoka, Akihiro Kishimura, Koichiro Kato, Michiko Furuta, Yoshihisa Yamashita

**Affiliations:** 1Graduate School of Systems Life Sciences, Kyushu University, Fukuoka 819-0395, Japan; 2Nissan Chemical Corporation, Tokyo 103-6119, Japan; 3Department of Applied Chemistry, Faculty of Engineering, Kyushu University, Fukuoka 819-0395, Japan; 4Manno-Cho Kokumin Kenko Hoken Soda Dental Clinic, Takamatsu 766-0201, Japan; 5Center for Future Chemistry, Kyushu University, Fukuoka 819-0395, Japan; 6Center for Molecular Systems, Kyushu University, Fukuoka 819-0395, Japan; 7Section of Preventive and Public Health Dentistry, Division of Oral Health, Growth and Development, Faculty of Dental Science, Kyushu University, Fukuoka 812-8582, Japan

**Keywords:** frailty, tooth loss, dietary problems, community-dwelling older people

## Abstract

This study aimed to investigate the association between dietary problems and frailty according to tooth loss in older Japanese people. This cross-sectional study included 160 older people (mean age 82.6 years) from Japan. Frailty status was assessed using the Study of Osteoporotic Fractures (SOF) criteria, which consists of (i) weight loss > 5% in the past year, (ii) inability to perform five chair stands, and (iii) self-perceived reduced energy level. Frailty was defined as the presence of ≥2 items of SOF criteria. Multivariate logistic regression analyses were performed with frailty as the dependent variable and dietary problems as the independent variable, stratified according to having <20 teeth. Low appetite and no enjoyment of eating were associated with frailty after adjusting for covariates in participants with <20 teeth. Dietary problems, including low appetite, eating alone, and negative attitudes toward enjoyment of eating were associated with a self-perceived reduced energy level in participants with <20 teeth. However, this association was not observed in participants with ≥20 teeth. In older people with fewer teeth, dietary problems have been suggested to be associated with frailty. Therefore, it may be necessary to pay attention to dietary problems, especially in older people with tooth loss.

## 1. Introduction

Frailty is a commonly recognized geriatric syndrome with a clinical state of increased vulnerability [1]. Fried’s phenotype model, which has been used abundantly in the literature to measure frailty, defines frailty as having three or more of the following factors: shrinking, weakness, exhaustion, slowness, and a low activity level [2,3,4]. Frailty is associated with a number of adverse health conditions, such as disability, dependency, cognitive impairment, and mortality [5]. To prolong a healthy life span, the prevention of frailty is gaining importance in aging societies such as Japan [6].

Dietary factors are considered to be closely linked with frailty [7]. Insufficient dietary intake leads to weight loss and sarcopenia (i.e., the age-related progressive loss of both muscle mass and muscle function), which may cause low muscle strength and a feeling of exhaustion [8]. Different aspects of dietary problems, such as loss of appetite, lack of enjoyment of eating, and eating alone contribute to decreased food intake [9,10,11], and therefore seem to be involved in the development of frailty. These dietary factors that cause reduced food intake should be considered to prevent frailty.

In older people, oral conditions are linked to appetite, enjoyment of eating and eating together [12,13,14,15]. Poor oral health may induce to select inappropriate food and food texture depended on oral condition, which leads to the loss of pleasure eating and reduction of appetite [14,15,16,17]. Maintaining oral condition may play an important role in maintaining enjoyment in meals and motivation for eating out and participating in a conversation [13]. This might contribute to increasing the willingness to eat together outside the household. Additionally, previous studies have reported that tooth loss is associated with frailty [18,19]. From this perspective, oral health is related to not only dietary factors but also to frailty. However, few studies have investigated the interrelationships between oral conditions, dietary factors, and frailty.

In this study, we hypothesized that dietary problems significantly affect frailty in individuals with more tooth loss. To verify this hypothesis, we investigated the association between dietary problems and frailty according to tooth loss in older adults.

## 2. Methods

### 2.1. Study Design and Study Population

This was a cross-sectional study conducted in the Kotonami District of Manno Town, Kagawa Prefecture, Japan, between 2017 and 2018. The Kotonami district is an underpopulated area with a total population of 2183, and the proportion of the older population aged 65 and older is 48.2%. Dental examination and questionnaire survey had been planned to include all community-dwelling older people who aged 75 and older and lived at home in the Kotonami District and to exclude those who were admitted to nursing care facilities. We performed the dental examination and questionnaire survey when they participated in the senior club, neighborhood community association, and hobby club, or received preventive or day service in the long-term care insurance system. A dental survey was conducted regardless of the presence or absence of tooth and prosthesis. A total of 252 community-dwelling older people (seven individuals aged 65–74 years and 245 individuals aged ≥75 years, mean age 82.4 years, 83 males and 169 females) voluntarily received the dental and questionnaire survey. The participation rate of the entire population aged ≥75 years was 46.8%. Of these, we excluded 92 participants who had missing data for oral assessments, information on participant characteristics, frailty status, or dietary problems. The total number of participants included in the analysis was 160 (mean age 82.6 years, 44 males and 116 females). This study was approved by the Kyushu University Institutional Review Board for Clinical Research (Approval No. 29-29). Written informed consent was obtained from all participants.

### 2.2. Measurements and Data Collection

Frailty status was assessed using the Study of Osteoporotic Fractures (SOF) criteria, which consists of three items: (i) weight loss > 5% in the past year, (ii) inability to perform five chair stands, and (iii) reduced self-perceived energy level (SE) [20]. Participants were considered “frail” if two or all of the components were present. The SOF index has been found to be as reliable and valid as the Cardiovascular Health Study (CHS) index, which is used the most globally. The SOF index has been reported to be simpler and more comparable to the CHS index [21]. SE was evaluated using the question “Do you feel full of energy?” as in a previous report [22]. Participants who responded “no” to the question were defined as reduced SE, and those who answered “yes” were defined as the non-SE group.

Oral assessments were performed by a dentist who recorded the number of teeth present.

To investigate the dietary factors of older people, the questionnaire on “appetite”, “eating alone”, and “attitudes toward enjoyment of eating” was used. Appetite was assessed via the Council on Nutrition Appetite Questionnaire (CNAQ) [23]. The CNAQ is an eight-item questionnaire, each rated on a five-point scale, and the total score can range between a minimum of eight and a maximum of 40 points [23]. Lower scores indicate deterioration in appetite, and a total score of ≤28 is defined as “loss of appetite” and has been shown to predict weight loss of at least 5% over six months in community-dwelling adults [23]. Eating alone was assessed by a single question item [24] as follows: “Do you often eat alone?” Attitudes toward enjoyment of eating were assessed using multiple question items [25], examples of which include the following: “Do you always enjoy your meals?” “Is it important to eat delicious food on weekdays?” “Is it important to eat delicious food on weekends?” “Do you always try to eat delicious food on weekdays?” “Do you always try to eat delicious food on weekends?”

Other variables used in the analysis included age, sex (male or female), education (≤9, 10–12, or ≥13 years), marital status (married, single/never married), family (living alone or not living alone), care situation (independent, requiring assistance, or requiring long-term care), ability to fill out forms by oneself (yes or no), smoking (never, past, or current), and alcohol consumption (number of drinks per day ≥ 3 or <3), mini-cognitive assessment (Mini-Cog) instrument (a total score of >3 or ≤2), medication (number of medications per day ≥3 or <3), and Charlson comorbidity index (CCI). The Mini-Cog consists of two components: a three-word recall task and the clock drawing test, and the total score can range between a minimum of zero and a maximum of five points. A total score ≤ 2 is defined as cognitive decline [26]. The CCI index adjusts for myocardial infarction, congestive heart failure, peripheral vascular disease, cerebrovascular disease, dementia, chronic pulmonary disease, connective tissue disease, peptic ulcer disease, mild liver disease, diabetes without end-organ damage, hemiplegia, moderate or severe renal disease, diabetes with end-organ damage, tumor without metastases, leukemia, lymphoma, moderate or severe liver disease, metastatic solid tumor, and acquired immunodeficiency syndrome (AIDS). The CCI score assigns a weight of 1–6 points to each comorbid disease, with the sum of individual scores serving as a measure of the overall comorbidity of a participant [27].

### 2.3. Statistical Analysis

The distribution of all variables included in the analysis, namely age, sex, education, marital status, family, care situation, ability to fill out forms by oneself, smoking, and alcohol consumption, Mini-Cog, medications, and CCI score was assessed. In addition to the descriptive analysis of associated predictor and outcome variables, a comparison analysis was also used to study statistical significance. Participants were divided into two groups: <20 teeth and ≥20 teeth. Previous studies have reported that the presence of at least 20 teeth was sufficient to ensure sufficient masticatory function, aesthetic satisfaction [28], and better oral health-related quality of life, which determines peoples’ perceptions of their oral health [29]. Differences between categorical groups were examined using the chi-square or Fisher’s exact test. Fisher’s exact test was used if the assumptions for the chi-square test were not met. Because the Shapiro-Wilk test revealed that all continuous variables (age, number of teeth present, and CCI) were not normally distributed, the Mann-Whitney U test was used to compare groups with continuous variables. Continuous variables were reported as median and interquartile range (25–75%), while categorical variables were reported as numbers and percentages. A logistic regression analysis was used to determine whether dietary problems (loss of appetite, eating alone, and attitudes toward enjoyment of eating) were associated with frailty in participants with or without <20 teeth, and then adjusted odds ratios (OR) and corresponding 95% confidence intervals (CI) were determined. To avoid multicollinearity effects, we performed Pearson’s correlation tests and discarded highly correlated variables (r > 0.7). There was a strong negative correlation between sex (female) and smoking status. Significant questions on attitudes toward the enjoyment of eating in the bivariate analysis were included in the logistic regression model. In the adjusted model, the covariates included age, care situation, ability to fill out forms by oneself, medication, and the CCI, which were significantly associated with frailty (Appendix A). R (version 4.1.2) was used for all analyses, and statistical significance was set at *p* < 0.05.

## 3. Results

The frailty, dietary problems, and associated factors of all participants (44 male and 116 female) and those with ≥20 teeth (20 male and 43 female) and <20 teeth (24 male and 73 female) are shown in Table 1. Each variable was significantly affected by the number of teeth present, with the exception of low appetite, sex, smoking, and alcohol consumption. The proportion of frailty status, excluding reduced SE, was significantly higher in the <20 teeth group. The frailty score was higher in the <20 teeth group (Appendix A).

When we examined the association between dietary problems and frailty in the ≥20 teeth group and <20 teeth group, there was no difference in appetite, eating alone, and attitudes toward enjoyment of eating between the no frailty and frailty groups in the ≥20 teeth group (Table 2 and Table 3). In the <20 teeth group, participants with low appetite had a higher percentage of frailty, weight loss, and reduced SE than those with high appetite (Table 2). The percentage of reduced SE was higher in participants who ate alone in the <20 teeth group than in those who ate together (Table 2). In the <20 teeth group, participants who answered “no” to the question “Do you always enjoy your meals?” and “Do you always try to eat delicious food on weekdays?” had higher percentages of frailty and reduced SE, respectively (Table 3). The results of the logistic regression analysis are presented in Table 4. In the <20 teeth group, low appetite was associated with frailty (OR = 3.84, 95% CI; 1.27–11.64) and no enjoyment of eating (OR = 13.50, 95% CI: 1.12–162.90). Additionally, low appetite (OR = 3.95, 95% CI: 1.03–15.10), eating alone (OR = 3.48, 95% CI: 1.37–8.83), and negative attitude toward enjoyment of eating (no eating delicious food on weekdays) (OR = 6.34, 95% CI: 1.63–24.67) were associated with reduced SE.

## 4. Discussion

Our study found that in the <20 teeth group, low appetite and no enjoyment of eating were associated with frailty, and dietary problems, including low appetite, eating alone, and negative attitude toward enjoyment of eating were associated with reduced SE as a frailty criterion. Reduced SE or exhaustion in older people is characterized by unusual fatigue or a general loss of energy. Previous studies have reported that exhaustion predicts the onset of functional limitations [30], disability [31,32], use of social and health services [33], and mortality [31,32]. Considering that improved SE is associated with improvements in frailty [34], the results of this study may be useful in formulating a strategy for extending the healthy life span of older people.

The finding that dietary problems were associated with frailty in participants with <20 teeth but not in those with ≥20 teeth suggests that chewing ability can avoid the effect of dietary problems on frailty. Even though participants with ≥20 teeth have dietary problems, high chewing ability can maintain food intake and contribute to normal nutritional status; subsequently, it may prevent frailty. Indeed, having more teeth (≥20 teeth) is associated with a lower chance of weight loss [35,36], frailty [18,19], disability [37,38], and mortality [37] in older people. Thus, maintaining oral health may prevent deterioration. Considering that approximately 80% of participants with <20 teeth used dentures (Appendix A), dietary problems could affect frailty, even when wearing dentures. Thus, denture use may not be sufficient to improve dietary problems and prevent frailty due to inefficient chewing ability. Efforts to retain natural teeth, such as the prevention of dental disease from earlier years, are important in preventing frailty in older people.

An important factor associated with malnutrition is poor appetite. It has been reported that loss of appetite is associated with a decrease in daily vitality using an active scale in older people [39], and our study showed an association between poor appetite and frailty. The regulation of appetite is complex and not completely understood. The effect of loss of teeth on appetite is controversial, with some reports suggesting that reduced chewing ability due to a reduced number of teeth present affects the enjoyment of food and appetite [40,41], and others suggesting that oral status is not related to appetite [42]. Although the reason for the association between a loss of appetite and frailty only in the <20 teeth group is unclear, to prevent frailty in older people, an approach aimed at improving appetite in addition to maintaining the remaining teeth is effective. Appetite is strongly influenced by the environment and mood, and eating alone is associated with poor appetite among older people [43,44]. Therefore, the use of congregate meal services, an opportunity to eat together with others, can be a measure to cope with exhaustion in older people associated with improved appetite.

Eating alone is associated with frailty [45] and depressive symptoms [46] in Japanese community-dwelling older people. The percentage of eating alone was higher in the <20 teeth group (54.6%) than in the ≥20 teeth group (34.9%) (Table 1), and eating alone was associated with reduced SE in the <20 teeth group (Table 4). In addition, people living alone were more likely to have the opportunity to eat alone [45]. The percentage of those living alone in the <20 teeth group (36.1%) was significantly higher than that in the ≥20 teeth group (19.0%) (Table 1). According to the survey in Kochi Prefecture, which defined eating alone in the same method as this study, the percentage of eating alone was reported to be 33.2% of the participants [24]. Compared to this study, the proportion of eating alone (46.9%) (Table 1) in our study was high. Comparing the percentage of participants who lived alone in each study, the rate was higher in our study (29.4%) (Table 1) than in the Kochi study (18.6%), suggesting that living alone is a background factor for eating alone. On the other hand, in community-dwelling older people, those who live with their families yet eat alone were found to be at risk for depressive symptoms [46]. With a rapidly aging society, eating alone may affect the quality of life among the older population. Strategies to keep older people from eating alone that consider psychosocial aspects should be considered.

This study had several limitations. First, because this was a cross-sectional study, it was not possible to estimate the causal relationship between frailty and related factors. Second, there is a possibility that different results may be obtained when different frailty assessment methods are used. In addition, we did not collect data on other factors that may affect frailty, and an unmeasured confounding bias may exist. Third, other dietary factors, such as skipping meals or night eating, have not been examined. Moreover, although appetite was assessed using standardized questionnaires, other dietary factors were assessed using a single or a series of simple questions, which may not be validated and reliable. Finally, the target area of this research, the Kotonami District of Manno Town, is designated as an underpopulated area by the Act on Special Measures for Promotion for Independence for Underpopulated Areas in Japan, and the study population (n = 160) was a convenience sample based on voluntary participation. This study was conducted between 2017 and 2018, and its results may not be considered up to date. In addition, low appetite, sex, smoking, and alcohol consumption was not significantly affected by the number of teeth present (Table 1). This study population was mostly comprised of women (72.5%) (Table 1). In general, women do not smoke and drink alcohol as much as men do [47]. A high percentage of women and low percentage of participants with smoking and drinking may have contributed to the lack of association with the number of teeth present. Therefore, the generalizability of the results is low, and similar studies with a larger sample size are needed. However, we believe that it is important to identify dietary problems as factors related to frailty in older adults in underpopulated areas.

## 5. Conclusions

Our study showed that dietary problems were associated with frailty in community-dwelling older adults with fewer teeth. The prevention of frailty and extension of the healthy life expectancy of older people needs to be considered by establishing both a professional collaboration system at the individual level and a support system at the community level. To improve frailty in older people, it may be useful to make efforts to preserve the remaining teeth at the individual level and, at the same time, to solve dietary problems and implement measures to cope with eating alone at the community level.

## Figures and Tables

**Table 1 ijerph-19-16260-t001:** Characteristics of the participants in the ≥20 teeth and <20 teeth group.

	All Participants	≥20 Teeth	<20 Teeth	*p*-Value ^a^
	n = 160	n = 63	n = 97	
Frailty	26.9%	15.9%	34.0%	0.011 ^b^
Frailty items				
Weight loss	17.5%	7.9%	23.7%	0.010 ^b^
Inability to do five chair stands	21.9%	9.5%	29.9%	0.002 ^b^
Reduced SE	68.1%	69.8%	67.0%	0.707 ^b^
Dietary problem				
Low appetite	22.5%	20.6%	23.7%	0.702 ^b^
Eating alone	46.9%	34.9%	54.6%	0.016 ^b^
Number of teeth present (median (25–75%))	16 (2–22)	24 (21–27)	3 (0–12)	<0.001 ^d^
Age (median (25–75%))	82 (79–86)	79 (76–83)	84 (80–88)	<0.001 ^d^
Female	72.5%	68.3%	75.3%	0.332 ^b^
Education				
≤9 years	50.0%	33.3%	60.8%	<0.001 ^b^
10–12 years	33.1%	38.1%	29.9%	
≥13 years	16.9%	28.6%	9.3%	
Single/never married	54.4%	36.5%	66.0%	<0.001 ^b^
Living alone	29.4%	19.0%	36.1%	0.021 ^b^
Care situation				
Independent	66.2%	82.5%	55.6%	<0.001 ^b^
Requiring assistance	13.8%	11.1%	15.5%	
Requiring long-term care	20.0%	6.3%	28.9%	
No ability to fill out forms by oneself	39.4%	20.6%	51.5%	<0.001 ^b^
Smoking				
Never	80.0%	77.8%	81.4%	0.781 ^b^
Past	15.0%	15.9%	14.5%	
Current	5.0%	6.3%	4.1%	
High alcohol consumption (≥3 drinks per day)	5.0%	6.3%	4.1%	0.528 ^b^
Cognitive decline *^1^	18.4%	4.8%	38.5%	<0.001 ^c^
Medication (≥3 per day)	74.4%	38.1%	82.5%	<0.001 ^b^
CCI (median (25–75%))	0 (0–1)	0 (0–1)	1 (0–2)	0.005 ^d^

^a^ Difference between the ≥20 teeth and <20 teeth group; ^b^ P for the chi-square test; ^c^ P for the Fisher’s exact test; ^d^ P for the Mann-Whitney U test; *^1^ Excluding participants with missing value (n = 8); SE, self-perceived energy level.; CCI, Charlson comorbidity index.

**Table 2 ijerph-19-16260-t002:** The association of low appetite and eating alone with frailty according to the number of teeth present.

	n	Frailty	*p*-Value	Frailty Items					
				Weight Loss	*p*-Value	Inability to Do Five Chair Stands	*p*-Value	Reduced SE	*p*-Value
*<20 teeth*									
Appetite									
High	74	28.4%	0.035 ^a^	17.6%	0.011 ^a^	29.7%	0.949 ^a^	60.8%	0.002 ^a^
Low	23	52.2%		43.5%		30.4%		87.0%	
Eating alone									
No	44	31.8%	0.677 ^a^	25.0%	0.786 ^a^	25.0%	0.337 ^a^	52.3%	0.005 ^a^
Yes	53	35.8%		22.6%		34.0%		79.2%	
*≥20 teeth*									
Appetite									
High	50	12.0%	0.194 ^b^	8.0%	1.00 ^b^	6.0%	0.096 ^b^	66.0%	0.311 ^b^
Low	13	30.8%		7.7%		23.1%		84.6%	
Eating alone									
No	41	12.2%	0.299 ^b^	9.8%	0.650 ^b^	4.9%	0.171 ^b^	63.4%	0.159 ^b^
Yes	22	22.7%		4.5%		18.2%		81.8%	

^a^ Chi-square test; ^b^ Fisher’s exact test; SE, self-perceived energy level.

**Table 3 ijerph-19-16260-t003:** The association between attitudes toward enjoyment of eating and frailty according to the number of teeth present.

	n	Frailty	*p*-Value	Frailty Items					
				Weight Loss	*p*-Value	Inability to Do Five Chair Stands	*p*-Value	Reduced SE	*p*-Value
*<20 teeth*									
Do you always enjoy your meals?									
Yes	92	31.5%	0.044 ^b^	21.7%	0.085 ^b^	30.4%	1.00 ^b^	65.2%	0.168 ^b^
No	5	80.0%		60.0%		20.0%		100.0%	
Is it important to eat delicious food on weekdays?									
Yes	86	33.7%	0.862 ^a^	22.1%	0.295 ^a^	31.4%	0.497 ^b^	65.1%	0.329 ^b^
No	11	36.4%		36.4%		18.2%		81.8%	
Is it important to eat delicious food on weekends?									
Yes	73	37.0%	0.282 ^a^	23.3%	0.864 ^a^	37.0%	0.009 ^b^	65.8%	0.646 ^a^
No	24	25.0%		25.0%		8.3%		70.8%	
Do you always try to eat delicious food on weekdays?									
Yes	71	35.2%	0.683 ^a^	23.9%	0.929 ^a^	32.4%	0.375 ^a^	59.2%	0.007 ^b^
No	26	30.8%		23.1%		23.1%		88.5%	
Do you always try to eat delicious food on weekends?									
Yes	60	33.3%	0.856 ^a^	21.7%	0.547 ^a^	35.0%	0.162 ^a^	63.3%	0.327 ^a^
No	37	35.1%		27.0%		21.6%		73.0%	
*≥20 teeth*									
Do you always enjoy your meals?									
Yes	58	17.2%	0.583 ^b^	8.6%	1.00 ^b^	10.3%	1.00 ^b^	67.2%	0.311 ^b^
No	5	0.0%		0.0%		0.0%		100.0%	
Is it important to eat delicious food on weekdays?									
Yes	58	17.2%	0.583 ^b^	8.6%	1.00 ^b^	10.3%	1.00 ^b^	72.4%	0.156 ^b^
No	5	0.0%		0.0%		0.0%		40.0%	
Is it important to eat delicious food on weekends?									
Yes	35	17.1%	1.00 ^b^	8.6%	1.00 ^b^	11.4%	0.684 ^b^	74.3%	0.390 ^a^
No	28	14.3%		7.1%		7.1%		64.3%	
Do you always try to eat delicious food on weekdays?									
Yes	43	18.6%	0.481 ^b^	7.0%	0.649 ^b^	11.6%	0.404 ^b^	69.8%	0.985 ^a^
No	20	10.0%		10.0%		5.0%		70.0%	
Do you always try to eat delicious food on weekends?									
Yes	27	18.5%	0.733 ^b^	7.4%	1.00 ^b^	11.1%	1.00 ^b^	74.1%	0.526 ^a^
No	36	13.9%		8.3%		8.3%		66.7%	

^a^ Chi-square test; ^b^ Fisher’s exact test; SE, self-perceived energy level.

**Table 4 ijerph-19-16260-t004:** Adjusted odds ratio of frailty for each dietary problem according to the number of teeth present.

		Frailty Items		
	Frailty	Weight Loss	Inability to Do Five Chair Stands	Reduced SE
	Adjusted OR (95% CI)			
*<20 teeth*				
Low appetite (ref. high)	3.84 (1.27–11.64) *	4.34 (1.38–13.66) *	1.33 (0.43–4.17)	3.95 (1.03–15.10) *
Eating alone (ref. no)	0.98 (0.40–2.45)	0.66 (0.24–1.79)	1.31 (0.49–3.48)	3.48 (1.37–8.83) *
Attitudes toward enjoyment of eating				
Do you always enjoy your meals? (ref. yes)	13.50 (1.12–162.90) *	7.03 (0.90–54.91)	0.70 (0.07–7.51)	N/A ^†^
Is it important to eat delicious food on weekends? (ref. yes)	0.49 (0.15–1.58)	1.00 (0.31–3.30)	0.98 (0.04–1.96)	1.36 (0.47–3.97)
Do you always try to eat delicious food on weekdays? (ref. yes)	1.07 (0.38–2.97)	1.21 (0.40–3.70)	0.74 (0.24–2.28)	6.34 (1.63–24.67) *
*≥20 teeth*				
Low appetite (ref. high)	2.42 (0.36–16.17)	1.89 (0.11–32.64)	3.36 (0.28–40.58)	2.60 (0.43–15.87)
Eating alone (ref. no)	0.77 (0.12–4.75)	0.16 (0.01–3.53)	2.16 (0.23–20.37)	1.87 (0.49–7.20)

A multivariate logistic regression model included frailty as the dependent variable and dietary problems as the independent variable. Independent variables were entered separately. Adjusted for age, care situation, ability to fill out forms, medication, and Charlson comorbidity index score. * *p* < 0.05; ^†^ Infinite OR estimates were achieved by data separation; OR, Odds ratio; CI, confidence interval; SOF, Study of Osteoporotic Fractures; SE, self-perceived energy level; N/A, not applicable.

## Data Availability

The data are not publicly available due to ethical restriction.

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
