# Peer review of "Dietary Problems Are Associated with Frailty Status in Older People with Fewer Teeth in Japan"

_ijerph, 2022, doi:10.3390/ijerph192316260_

Round 1

Reviewer 1 Report

Dear All,

The study tackles a very significant, although often overlooked problem affecting older people.

 While revising the paper the following issues need to be addressed:

1.The authors should supply information on whether the cognitive state of the respondents was assessed. It is a factor that might have significantly affected the obtained results, especially those that were obtained from the questionnaires.

2.The information should be provided whether the study has received approval of the relevant bioethics committee or what the research method was.

3.  It might be judicious to refrain from referencing certain works, especially those that have been published long time ago, unless they are crucial for the essence of the paper.

4. The time period when the study was conducted  raises doubts as to whether the results might be considered up to date.

Author Response

We would like to thank the editor and reviewers for their detailed review and insightful comments. We have made every effort to revise the manuscript based on the reviewers’ recommendations and suggestions.

  • Please see the attachment for the reply.

Reviewer 2 Report

I enjoyed reading this paper. The authors made a good effort to provide data on the correlation between dietary problems and frailty and oral health in a Japanese population. A few comments for consideration by authors:

INTRODUCATION

1.    Authors need to provide a clear hypothesis in Introduction.

2.    There are many dietary problems. Why did you only select appetite, eating alone and attitudes toward enjoyment of eating?

METHODS

3.    The study sample size (160) is not large. How did author get this sample size?Is this sample size sufficient or appropriate for the study?

4.    It is good for taking sex as a confounding factor in your analyses. However, it seems to lack of information about diseases and medication that could be an important information for the studies in older people.

5.    What are the criteria for sample inclusion or exclusion 

6.    The author evaluate appetite using a standard questionnaire, but for other dietary factors, they only used single or several simple questions. I just wonder about the reliability and validity about these questions.

7.    For example, if the questions about eat alone was developed by the author, but not cited from other standard questionnaire or other study, it may be better to add a time period for this question, such as “Do you often eat alone in the past month?”, because dietary habit could be changed.

RESULTS

8.    Why are the positions of Table 3 and Table 4 reversed?

DISCUSSION

9.    Limitation part could be improved. There are some limitations that you did not mention. For example, other dietary factors, such as skipping meals or night eating, were not examined in the study. And the assessment of eat alone or enjoy the meal is not validated (if so).

Author Response

(The authors gave the same response as above.)

Reviewer 3 Report

1.- Inclusion criteria have to be well defined.

2.- Why have you choose "20 teeth" as limit? 

3.- You have written: "Each variable was significantly af-125 fected by the number of teeth, with the exception of low appetite, sex, smoking, " in the results section. Nevertheless, many authors have studied this relation. Include it and justify your results in the discussion section. 
4.- Conclusions must be rewritten according the previous suggestions.

Author Response

(The authors gave the same response as above.)

Round 2

Reviewer 1 Report

Dear All,

The authors took into account previous suggestions.

Author Response

Thank you for the comments.

Reviewer 2 Report

The author answered all my comments. This article could be accepted in current form.

Author Response

Thank you for the comments.

Reviewer 3 Report

Dear authors, 

Reviewed version has substantially improved the initial one.  

Anyway, the inclusion and exclusion criteria are not well defined. You talk about the number of teeth, but... have you consider the presence of any prosthesis? For example.

Have you calculated the sample size before the study? Why? How?

Kind regards

Author Response

We thank the reviewer for the careful consideration. We had not calculated the sample size before the survey because we had been planned the survey including all residents who aged 75 and older and lived at home in Kotonami district of Manno Town. We have revised the manuscript to clearly describe the inclusion and exclusion criteria as follows: “Dental and questionnaire survey had been planned to include all community-dwelling older people who aged 75 and older and lived at home in the Kotonami District and to exclude those who admitted to nursing care facilities. We performed the dental and questionnaire survey when they participated in the senior club, neighborhood community association, and hobby club, or received preventive or day service in the long-term care insurance system. Dental survey was conducted regardless of the presence or absence of tooth and prosthesis. A total of 252 community-dwelling older people (7 individuals aged 65-74 years and 245 individuals aged ≥ 75 years, mean age 82.4 years, 83 males and 169 females) voluntarily received the dental and questionnaire survey. The participation rate of the entire population aged ≥ 75 years was 46.8%.” (METHODS, lines 66-75, p2)

          We had mentioned denture wearing in the Discussion section according to the academic editor’s suggestion in the previous revision as follows: “Considering that approximately 80% of participants with < 20 teeth used dentures (Supplementary Table 3), dietary problems could affect frailty, even when wearing dentures. Thus, dentures use may not be sufficient to improve dietary problems and prevent frailty due to inefficient chewing ability. Efforts to retain natural teeth, such as the prevention of dental disease from earlier years, are important in preventing frailty in older people.” (DISCUSSION, lines 196-200, p8)